# Disorders of the Female Reproductive Tract in Chelonians: A Review

**DOI:** 10.3390/ani15091275

**Published:** 2025-04-30

**Authors:** Emanuele Lubian, Giulia Palotti, Francesco Di Ianni, Alessandro Vetere

**Affiliations:** 1Department of Veterinary Medicine, University of Milan, via Dell’Università 6, 26900 Lodi, Italy; giulia.palotti@guest.unimi.it; 2Mypetclinic, Viale Ranzoni, 20149 Milano, Italy; 3Department of Veterinary Science, University of Parma, Strada del Taglio, 10, 43126 Parma, Italy; francesco.diianni@unipr.it (F.D.I.); alessandro.vetere@unipr.it (A.V.)

**Keywords:** chelonian reproduction, reproductive disease, female reproductive tract, dystocia, ectopic eggs, follicular stasis

## Abstract

Knowledge of reproductive biology and related disorders is fundamental in the clinical management of captive chelonians. This literature review provides an overview of the main pathological conditions that afflict the female genital tract of these animals; for each condition, the practical and effective diagnostic and therapeutic procedures are described. The reported disorders are dystocia, ectopic eggs, follicular stasis, infertility, oophoritis, salpingitis, cloacitis, cloacal/oviductal prolapse, tumors, and ovarian torsion. A thorough clinical examination, associated with an accurate medical history collection and diagnostic tests, such as ultrasound, radiography, CT, and endoscopy, is essential for making an accurate diagnosis. Surgery is often required if medical therapy proves to be unsuccessful.

## 1. Introduction

The number of exotic pets, including tortoises, presented to veterinary practitioners for clinical evaluation due to pathological conditions or routine health examinations is increasing.

Owners more and more often rely on veterinary surgeons to find the cause and treat the reproductive disorders that afflict their chelonians. As the presence of these disorders in captive animals has risen, so has the quality of veterinary treatments and diagnostic techniques for these diseases [1].

Reproductive disorders in chelonians, particularly in captive settings, are a growing concern, with certain pathologies being more prevalent than others. Among these, conditions such as follicular stasis, dystocia, and infertility are relatively common and are often linked to suboptimal environmental management. In particular, poor temperature regulation, inadequate humidity, and an improper photoperiod have been recognized as key factors contributing to these disorders. In comparison with other reptile taxa, chelonians tend to suffer from a higher frequency of reproductive issues, likely due to their long lifespan, delayed sexual maturity, and specialized environmental requirements. Additionally, chelonians are often housed in environments that do not fully replicate their natural habitats, leading to disruptions in their reproductive cycles.

Other conditions, such as ectopic eggs, oophoritis, and salpingitis, while less frequently encountered, are still significant and can be exacerbated by chronic stress, malnutrition, or poor management practices. In contrast, species like snakes and lizards may exhibit a lower incidence of these specific reproductive issues, likely due to differences in their reproductive strategies and more adaptable environmental tolerances. Chelonian reproductive health, therefore, appears more susceptible to environmental stressors and management-related factors, which are less of an issue in taxa with shorter lifespans and more robust reproductive adaptations [2,3,4,5].

Furthermore, reproductive disorders in chelonians may also be influenced by extra-genital factors. Systemic diseases, such as nutritional deficiencies, infections, or metabolic disorders, can negatively affect reproductive health. For example, calcium or vitamin D3 deficiencies are well-documented in reptiles and can lead to egg binding or difficulty in follicle development, predisposing females to dystocia or other complications. Environmental stressors, including poor habitat conditions and improper handling, can further exacerbate reproductive issues, making it crucial for veterinarians to consider these external factors when diagnosing and treating reproductive disorders in chelonians [2,3,4,5].

Herein, we review the pathological conditions affecting the reproductive tract of chelonians based on the existing scientific literature. For each condition, we aim to summarize practical and effective diagnostic and therapeutic procedures, considering basic anatomy, physiology, and endocrinology as essential prerequisites. The databases Google Scholar, PubMed, and Scopus were searched with the following keywords: “chelonian, reptile, reproduction, reproductive diseases, dystocia, follicular stasis, salpingitis, cloacitis, oviductal prolapse, neoplasia”. Only peer-reviewed articles were considered for inclusion. Articles that did not meet this criterion or were deemed irrelevant to the review’s objectives were discarded.

### 1.1. Anatomy of the Female Reproductive System

The female reproductive tract consists of a pair of ovaries, oviducts, and the cloaca.

The ovaries are located anterior to the kidneys and are suspended in the coelomic cavity by the mesovarium. Their sizes vary according to the season and the stage of oogenesis: during inactivity, they are small and granular in appearance; during the reproductive season, they consist of large clusters of yellow spherical follicles [6]. The ovarian vessels are located in the mesovarium [7].

The oviducts are two tubular structures extending from the ovaries to the cloaca, positioned ventrolaterally to the rectum. Each oviduct opens laterally into the cloaca through the genital papilla [7,8,9,10].

The cloaca is an organ with a tubular structure that opens with an orifice in the cranio-ventral portion of the tail. It is composed of three chambers, from the most cranial to the most caudal: the coprodeum, urodeum, and proctodeum. The coprodeum receives feces from the distal colon. The urodeum receives the terminal opening of the urogenital tract (ureters, oviducts, and urethra). The proctodeum contains a copulatory organ (phallus in males and clitoris in females) [11,12,13,14].

### 1.2. Reproductive Physiology and Endocrinology

Reptiles have a neuroendocrine regulation system for reproduction that is very similar to that of other vertebrates. Sensory systems and specific brain areas act by integrating internal and external stimuli, inducing the seasonal changes necessary for the reproductive system [13].

GnRH induces the release of gonadotropins (FSH and LH) from the anterior pituitary gland), influencing the gonadal development and function and inducing the secretion of steroid hormones (progesterone, testosterone, and estrogens). These hormones regulate hypothalamic and pituitary hormone secretion through a negative feedback mechanism [13,14].

Chelonians living in temperate climates follow a seasonal cycle, with follicular activity occurring only during the warmer seasons and egg deposition concentrated during the spring–summer period. In contrast, tropical species have different reproductive cycles, which are influenced by the dry and rainy seasons, with some species laying eggs during the dry season and others during the rainy season [12].

The ovarian cycle can be divided into four phases:Quiescent phase: there is no follicular development in the ovaries due to immaturity or hibernation.Folliculogenesis phase: The ovary matures and follicles develop under hormonal stimulation. The follicles can be distinguished as primordial, primary, secondary, or tertiary according to their developing stage [15]; in *Trachemys scripta* and *Trachemys venusta*, four classes of follicles were described and defined according to their diameter as follows: Class I (≤6 mm), Class II (7–13 mm), Class III (14–20 mm), and Class IV (≥21 mm) [16,17,18]. Under FSH stimulation, the follicles produce estradiol, with only some follicles ovulating while others undergo atresia [1].

Ovulation can be triggered by male presence, with male courtship behaviors (e.g., head butting) or mating, alongside the involvement of sexual pheromones [19]. Sexual behavior can also occur between two females or between females and males. In temperate regions, vitellogenesis typically begins in late August, reaching full follicular maturation in autumn before hibernation, allowing for immediate mating post-hibernation [20,21].

3.Fertilization and pregnancy: After ovulation, the oocyte and the sperm meet inside the oviduct, which secretes progesterone to maintain pregnancy. The yolk and eggshell form as the egg progresses, with subsequent shell calcification.4.Oviposition: Calcified eggs remain in the oviduct before being laid, with the duration varying by species; some species can retain them for up to 4–6 months [6,8]. This period may lengthen if environmental conditions are unfavorable for laying or if the animal has a concurrent pathological condition [6,8].

Females can store sperm in the cloaca, uterus, or infundibulum for extended periods (e.g., months to years). This ability has been observed in species from the families Emydidae [21], Kinosternidae [22], Chelydridae [22], Trionychidae [23], Cheloniidae [24], and Testudinidae [25]. Females can store sperm from multiple males, resulting in offspring with multiple paternities [22,26].

## 2. Clinical Examination

After having collected the data of the owner and of the animal, obtaining the anamnestic information of the animal is fundamental. After a general anamnesis, it is always good to carry out a problem-oriented anamnesis. Regarding the reproductive system, it is good practice to first know the species being examined and their reproductive habits in the wild; for example, some species lay one or two eggs at a time several times a year (e.g., *Platemys platycephala* [27] and *Kinosternon* sp. [28,29]), while others (e.g., Cheloniidae or Dermochelydae [30,31]) lay numerous eggs in a single deposition.

The clinical examination should not be focused only on reproductive problems; often, pathologies of the genitourinary apparatus are consequences (or manifestations) of problems concerning other organs or apparatuses; the opposite is also true, where reproductive pathologies can affect other apparatuses.

For particular observational examinations of the reproductive apparatus, observation and palpation are used. Visual assessment and digital exploration of the vent should be performed. It is also important to place a finger in the prefemoral fossa and gently swing the specimen; solid structures can be perceived caudally with the tip of the finger (calcified egg or stones) [32].

Collateral tests are useful in the course of reproductive pathology.

### 2.1. Collateral Tests

#### 2.1.1. Hematology and Biochemistry

Blood tests help clinicians make a proper diagnosis of a reproductive disease; however, there are numerous factors that can influence the results, such as the sampling site (lymphodilution), the environmental temperature, and stress [33]. Notably, physiologically, some blood values are modified during folliculogenesis and vitellogenesis, particularly triglyceride, cholesterol, and calcium levels [1,34], as well as the total protein and albumin levels [35].

#### 2.1.2. Blood Biochemistry Values

One of the most important parameters to evaluate is calcium. Concentrations of tCa (total calcium) and iCa (ionized calcium) certainly change according to the concentration of albumin, sex, and species; however, it has been demonstrated that in specimens with reproductive disorders, the concentration is greater (+0.40–0.68 mmol/L), so it is possible to establish a cut-off of 2.2 mmol/L (with a sensitivity of 81.8% and specificity of 76.4%) to distinguish who is affected or not affected by a reproductive disorder [36]. Several studies reported an increase in the serum calcium concentration during follicular stasis [34,35]; however, during reproductive pathology, calcium can also decrease, for example, as a consequence of the continuous deposition of calcium on chronically retained eggs [33,34,37].

Increases in LDH and AST levels have been reported in specimens with ovarian teratoma [38] and cloacal prolapse [32]. An increase in the serum ALB concentration, total protein concentration, and ALP concentration has been reported in specimens with follicular stasis [34,35]. Hypoglycemia can occur during septicemia (e.g., egg yolk coelomitis) [32].

#### 2.1.3. Hematological Findings

Anemia has been reported in chelonians with fibropapillomatosis [39,40] and follicular stasis [34]. Heterophilia, which can be indicative of infection and inflammation [32], has been revealed in cases of ovarian teratoma [38], granulomatous oophoritis, chronic egg retention [37], egg yolk coelomitis [34], and fibropapillomatosis [41]. Eosinopenia is reported during the course of parasitism [32]. Heteropenia is reported in follicular stasis [34]. Leukopenia is reported in follicular stasis [34] and fibropapillomatosis [39]. Monocytosis is reported in ovarian teratomas [38], granulomatous oophoritis [37], and fibropapillomatosis [42].

In conclusion, importantly, the usefulness of blood tests depends on the presence or absence of validated species-specific ranges [43].

#### 2.1.4. Cytology

Cytology is a quick and inexpensive exam that can help clinicians make a diagnosis. Reproductive tract disease is generally detected in the blood, coelomic and cloacal fluids, and tumors. Depending on the situation, it is possible to obtain samples with swabs, fine needle aspiration, or tissue imprinting, and generally, normal cellularity or inflammation, infection, foreign material, and cancer can be detected [44].

Inflammation can be classified as heterophilic, eosinophilic, mixed cell, or macrophagic, and the possible cells are heterophils, eosinophils, lymphocytes, plasma cells, and macrophages; the type of cells may indicate the etiology and pathogenesis [44]. Acute inflammation is characterized by a number of heterophiles greater than 75%, and when the heterophiles are toxic (i.e., with degranulation and vacuolization), the inflammatory response is considered more aggressive. On the other hand, during chronic inflammation, it is possible to observe a mixture of lymphocytes, macrophages, and heterophiles. The presence of epithelioid cells and fibroblasts is indicative of granuloma [44].

Hyperplasia and benign neoplasia are impossible to distinguish, and the criteria for diagnosing malignant neoplasia include cellular, nuclear, cytoplasmic, and structural features [44].

The presence of pathogenic agents, such as bacteria, fungi, or mycobacteria, can be revealed by the use of specific stains (gram, blue methylene, and Ziehl Nielsen, respectively). Even without specific stains, foreign bodies, e.g., droplets of yolk in coelomic fluids, can be observed during egg coelomitis [32].

In conclusion, this method tends to produce false negative results; thus, a biopsy and histopathology should be considered additional exams [45].

#### 2.1.5. Histopathology

Histology makes it possible to quickly identify a tissue, an organ, or a pathological condition of both whose exact nature is not known. The conventional method of a biopsy is to place the sample in a 10% solution of formalin (10:1 ratio) and collect more than one sample from the same tissue [46]. The main reproductive pathology for which histology is required is tumors.

#### 2.1.6. Parasitology

Fecal samples are often valuable diagnostic tools for sick reptiles. It is important to identify both endoparasites and ectoparasites, both of which can influence an animal’s general health and coexist with reproductive pathology. Sometimes they can be considered causes of cloacitis, or they can appear as a consequence of a reproductive pathology (myiasis during cloacitis). Ectoparasites may be collected in 70% ethanol and then submitted to the laboratory for identification [46].

#### 2.1.7. Microbiology and Virology

It is important to culture aerobic and anaerobic bacteria and to perform antibiotic susceptibility testing, but first, knowledge of the normal microflora of chelonians is important for interpreting microbiology results [47,48,49,50,51].

In the oviductal fluid of healthy *Chelonia mydas*, during and before egg laying, seven genera of bacteria can be isolated, most of which could be pathogens [51].

In the cloacal tract, many bacteria have been found in animals without clinical signs [47,48,49,50,51,52], and only some of them can be considered pathogens [50]. Additionally, molds can be isolated from the cloacal tract of asymptomatic patients [47].

The habitat and the alimentation influence the cloacal microbiota [50], as do the age of the subject (difference between juvenile and adult [49]) and the timing of exams (e.g., before or after egg laying [51]).

In conclusion, the discovery of potential pathogenic bacteria is not synonymous with disease and should be carefully evaluated; therefore, the authors recommend that a histological examination be performed when possible to confirm that the isolated bacterium is the cause of the pathology.

Viral infection may be suspected by histopathological or cytological findings [32]. Samples for viral isolation include excreted material, body fluids [e.g., blood, coelomic fluid, or cloacal discharge], dry swabs of lesions or luminal surfaces, or tissue biopsies [51]. It is important to freeze the sample (approximately −5 °C) to minimize the risk of bacterial contamination; additionally, when performing cloacal swabs, fecal contamination can denature any present viruses [51]. DNA can be extracted from the sample for PCR-based isolation of the viral genome (e.g., ChHV5 DNA during fibropapillomatosis) [41].

### 2.2. Diagnostic Imaging of the Female Reproductive Apparatus

#### 2.2.1. Ultrasound

The acoustic window is the prefemoral fossa, which allows for the visualization of the kidney, bladder, reproductive tract, coelom, and eggs [33].

Follicles are often visible immediately caudal to the liver in adult females [33]; they are visualized as spherical, homogeneous, echodense, and up to 25 mm in size in Mediterranean tortoises and up to 42 mm in giant species [53]; when they undergo atresia, they become small and hyperechoic until they disappear.

The oviduct is a thin, pleated, hyperechoic tubular structure that extends beyond the ovaries.

The eggs can be well visualized and located, and their contents can be analyzed; in particular, developing embryos can be observed, using the vertebral column as the most easily distinguishable feature [32].

Serial ultrasound examinations allow for the evaluation of the evolution of follicles and/or eggs.

#### 2.2.2. Radiology

Dorsoventral and lateral views provide sufficient information about the reproductive tract [33], but the dorsoventral view is the most common and rapid way to evaluate the presence of eggs (Figure 1).

The lack of coelomic fat between the organs makes the identification of ovaries, follicles, and oviducts difficult. Clinicians cannot visualize noncalcified eggs; however, this approach allows for excellent visualization of the shape and size of well-calcified eggs [54]. Superimposition artefacts are another disadvantage [54] (Figure 2).

#### 2.2.3. Computed Tomography

Sagittal, transverse, or horizontal scanning can be performed. Transverse scanning allows clinicians to evaluate the symmetry of paired organs [33] (Figure 3). It evaluates the entire genital system without overlapping artefacts; in particular, it provides detailed information on the number, size, shape, density, and position of follicles and eggs [23,55] according to its grayscale superiority to radiography [54].

The oviducts appear as hyperdense tubular structures in the caudal coelom [54] (Figure 4).

#### 2.2.4. Coelioscopy

Coelioscopy allows for direct visualization of the genital system and is very useful for identifying sex in young specimens.

Two approaches have been described: one for the prefemoral fossae (in lateral or dorsal recumbency) or one for plastronotomy; however, in both cases, the endoscope is introduced into the coelom to permit identification of the reproductive tract [56].

The ovary lies in the middle region of the coelom; however, in lateral recumbency, there is slight dorsal displacement [33]. It is described as small and granular in sexually inactive specimens and as cluster-shaped with an ochre-yellow color in mature adults. The inactive oviduct appears as a flattened, corrugated-looking, pale-cream-colored structure. If active, it appears distended and may contain eggs [33] (Figure 5).

#### 2.2.5. Cloacoscopy

This technique allows for direct visualization of the cloaca, the bladder, the oviducts, and the eggs. Since transparency of the bladder wall allows us to evaluate the gonads, this approach is not always feasible.

The oviducts can be explored during a cloacoscopy for a limited length [6]. The presence of accessory bladders allows the clinician to visualize the coelomic organs through the transparent wall (as well as in the urinary bladder) (Figure 6).

## 3. Main Medical Techniques

### 3.1. Oxytocin Therapy

Oxytocin is used to stimulate oviductal contractions and induce egg deposition [57]. Reptiles normally do not produce it (only mammals do), but an equivalent in reptiles is vasotocin [56]. Oxytocin, however, has been reported to be successful in 90% of cases when it is used within 48 h of problem onset [58,59].

Before using oxytocin, it is important to prepare a suitable egg-laying environment [60] and correct mineral and water imbalances to decrease the risk of a worsening underlying metabolic disease and/or rupture of a dehydrated friable oviduct [34]. Some authors believe that uterine tissue must be hydrated to ensure that oxytocin is effective; for all these reasons, it is appropriate to rehydrate the animal with a bath or active fluid therapy [60,61,62].

Temperature influences the effect of oxytocin on oviductal muscles; for this reason, the patient must be kept within the preferred temperature range for the species [34]. Lubrification of the cloaca is another important precaution [60].

Pretreatment with calcium has been reported to improve the contractility of the oviduct [60]. It is suggested to administer calcium gluconate 30 min before [63], one hour before [1], or several hours before [60].

There are different protocols for oxytocin treatment reported in the literature. The dosages are between 1 and 20 UI/kg [1,34,37,55,60,62,63,64,65], and low dosages are generally sufficient in Mediterranean tortoises [34]. Several studies have reported higher dosages (5–30 UI/kg [59] or 1–40 UI/kg [57]). There are different routes of administration: intraosseous [6,34,60], endovenous [6,66], intramuscular [1,6,34,37,58,59,60,66], and subcutaneous [63,67]. Endovenous administration results in more rapid induction [57,59] and does not cause side effects [57]. If a single administration of oxytocin does not yield a favorable outcome, the generally recommended protocol is to repeat the procedure a maximum of 3 times at a duration of 90 min with incremental doses or from 50% to 100% of the initial dose 1 to 12 h later [1,6,60,61].

There are possible complications associated with oxytocin administration, such as oviductal spasms and rupture and ectopic eggs in the urinary bladder [58,59,61,67,68,69,70].

The use of oxytocin in combination with prostaglandins, beta blockers (such as propranolol or atenolol) to enhance oxytocin activity, and medetomidine to relax sympathetic tone has also been reported [6,34]. The local application of prostaglandin E gel on the cloaca has been recommended, but there is no scientific confirmation of its beneficial or adverse effects [6,34].

### 3.2. Other Therapies

The controlled-release deslorelin implant is used in numerous species; few studies have been performed on turtles, and everyone has concluded that GnRH A-SRIs containing deslorelin cannot be considered good contraceptives [71,72,73,74].

Antibiotic therapy should always be preceded by an antibiogram test; however, in severe cases, it can be started empirically while waiting for the report. Anti-inflammatory drugs and antidolorific drugs are often used as therapies for reproductive diseases or for the management of surgical procedures.

## 4. Surgical Techniques

In chelonians, the decision between the surgical and medical management of reproductive disorders depends on the precise diagnosis; clinical status of the patient; the chronicity of the condition; and the presence of complications, such as yolk coelomitis or obstructive syndromes. When medical therapy fails or when the patient shows systemic signs (e.g., lethargy, inappetence, hyperuricemia), surgical intervention becomes necessary to avoid progression to more severe conditions [61,75].

Among surgical options, the prefemoral approach is considered the method of choice when accessible. It is less invasive, associated with shorter healing times, and carries fewer postoperative risks compared with a plastronotomy [56,76,77]. It is the preferred method for elective sterilization in chelonians (Figure 7). This approach is also especially useful in cases of chronic follicular stasis or in cases of dystocia where medical therapy has proven ineffective or inappropriate due to the risk of yolk leakage or inflammation [56,76,78] (Figure 8).

Conversely, a plastronotomy is reserved for cases where the prefemoral route is not feasible, such as in the presence of large or dystocic eggs, extensive fibrosis, calcified structures, or inaccessible contralateral reproductive organs [76,79,80]. Though more invasive and requiring a longer recovery period, plastronotomy allows for full access to the coelomic cavity and is often indispensable in advanced cases of yolk coelomitis or when precision is required in managing fragile or pathological tissues [76,81,82].

As an emergency or palliative measure, cloacal ovocentesis can be performed to relieve obstructive signs caused by retained or pathologically altered eggs. This technique is especially valuable when surgery is contraindicated or unaffordable [34,60,75]. Following egg aspiration, fragments are removed with caution to avoid injury to the oviduct or cloaca. However, this method is not curative and carries risks of postprocedural complications, including salpingitis and egg yolk coelomitis [37]. CT-guided ovocentesis has been reported in isolated cases, such as in *Terrapene carolina*, and may offer a less invasive alternative in specific clinical scenarios [83].

## 5. Diseases of the Reproductive System

### 5.1. Preovulatory Follicular Stasis (PFS)

Follicular stasis is a pathological condition in which ovarian follicles fail to ovulate or regress, leading to their prolonged retention within the ovaries; resulting in persistent follicles; and leading to inflammation and eventually rupture, coelomitis, and death [84,85,86] (Figure 9).

#### 5.1.1. Causes

The main causes are incorrect environmental management (irregular hibernation, alteration of the photoperiod and thermoperiod, incorrect dietary management) and the presence of concomitant pathologies (e.g., systemic pathologies, hormonal disorders, ovarian cysts, oophoritis, tumors) [38,61].

#### 5.1.2. Clinical Signs

Follicular stasis in chelonians can manifest as chronic anorexia, lethargy, abdominal pain, weight gain, hindlimb paresis, or absence of defecation and difficulty walking for several months [39,61,87]. Considering that this is a chronic pathology and that clinical signs can arise months after its appearance, there is no constant seasonality [39].

#### 5.1.3. Diagnosis

Blood tests: Sometimes there is hypercalcemia, hyperalbuminemia, elevated TP and ALP levels, moderate anemia/hemoconcentration, leukopenia, and heterophilia [6,8,34]. An increase in total calcium is considered indicative but not diagnostic [34,35,60]. In a recent survey on *Geochelone elegans*, no notable alteration was found [88].

Radiography: during follicular stasis, an increase in radiopacity can be noted in the middle of the coelom [61].

Serial ultrasound: Follicular evaluations performed with ultrasound scans repeated over time show no signs of progression or regression [18,36,37,39,53,61,77,89]. If the average number of follicles is high for the species, a stasis should be considered [1,39,61].

CT: This technique permits the evaluation of the number, size, shape, position, and radiodensity of follicles [56]. Pulmonary compression caused by follicles has been reported [88].

Cloaco-cystoscopy and exploratory celiotomy.

#### 5.1.4. Treatment

Treatment can be managerial, medical, or surgical, but the recommended treatment is an ovariectomy [38,39,88] (Figure 10). Spontaneous resolution of this pathology in chelonians has not yet been demonstrated [39].

### 5.2. Dystocia

Recognizing true dystocia from a physiological perspective is very difficult, primarily because it is not possible to exactly quantify the duration of gestation, i.e., the period between the detachment of ovules from the ovary and oviposition [38].

#### 5.2.1. Causes

The causes of dystocia can be classified into obstructive and nonobstructive [64,69,71].

Obstructive dystocia (egg binding) occurs when the eggs are not laid and retained inside the oviduct for an indefinite period of time or they are ectopic. Possible causes can be related to abnormal eggs (macrosomia, malformations) (Figure 11), egg adhesions to the oviduct mucosa, or maternal abnormalities. These, include a misshapen pelvis; oviductal sthenosis; or coelomic pathology, including cystic or cloacal calculi, constipation, organomegaly, abscesses, or neoplasia [38,66,68,86].

Nonobstructive causes are divided into two macrocategories: pathological nonobstructive causes (infectious states of the reproductive tract, nutritional deficiencies) and poor management nonobstructive causes (absence/inadequacy of the oviposition site, incorrect temperature/humidity/photoperiod, incorrect diet/malnutrition/obesity, dehydration [70], traumas/wounds/cloacitis, maladaptation syndrome in wild-captured animals, and gestation in females who have not mated for several years [38]).

Prolonged or complicated dystocia can lead to ectopic egg retention. When ectopic, it is instead possible to find the eggs free in the coelomic cavity (due to oviductal rupture or functional obstruction) [41,68]; in the colon; or, more frequently, in the urinary bladder [1,41,59,65,69,71]. Causes are an anatomical malformation [59,71], oxytocin-based therapies [59,60,62,65,69,70,71], previous salpingotomy without suturing of the thin-walled oviduct [60], and trauma (e.g., oviductal rupture).

#### 5.2.2. Clinical Signs

Distinguishing between dystocia and a normal reproductive cycle or gravidity presents significant challenges. The complexity of accurately diagnosing true dystocia as opposed to normal gravidity is particularly evident in chelonians. In these species, individuals may voluntarily retain eggs in the oviduct for up to six months, waiting for optimal conditions for oviposition [62,90].

In the early stages, clinical signs may be absent or nonspecific; sometimes, there is an increase in the animal’s restlessness, and continuous attempts to lay eggs occur [34].

In advanced or chronic forms, retained eggs may degenerate and cause salpingitis, with eventual rupture of the oviduct and consequent egg coelomitis or ectopic egg and/or compression of adjacent organs (bladder, colon, stomach, lungs) (Figure 12). Another clinical sign is paralysis of the hind limb due to compression of the obturator nerve [58,70]. The resulting clinical signs can include debilitation, anorexia, dehydration, abnormal posture, breathing difficulties, air hunger, inactivity, and malodorous cloacal loss [61]. Dilatation of the prefemoral fossa [37,85], limited movement and inappetence [68], cloacal bleeding [67], hyperactivity, and dyspnea [37] have been reported in the presence of ectopic eggs.

#### 5.2.3. Diagnosis

A thorough medical history, such as information on previous depositions and management, is the first step in making a correct diagnosis. During a clinical examination, the palpation of the prefemoral fossa sometimes allows the eggs to be perceived (it is impossible for clinicians to differentiate them from bladder stones).

Blood tests may reveal hypocalcemia (secondary nutritional hyperparathyroidism/sequestration by chronically retained eggs) [34]. Hyperkalemia and hyperuricemia have been reported in cases that involved ectopic eggs; however, often, the blood profile is not indicative [83].

Radiographic examination allows for the confirmation of the presence, number, shape, size, and content of eggs (recognizing the exact location is more difficult). Radiographic signs of chronic egg retention may include the excessive mineralization of eggshells (which are sometimes lamellar after a prolonged period of time) [37,58].

Ultrasound, computed tomography, and cystoscopy allow the clinician to locate the eggs exactly and/or evaluate other concomitant pathological conditions [58].

CT: This technique provides detailed information on the number, size, shape, density, and location of eggs [55].

#### 5.2.4. Treatment

If dystocia is caused by management errors, it is often sufficient to resolve them, and the animal lays eggs; however, this tends to be possible when the problem is identified quickly. Sometimes, however, it is necessary to resort to a pharmacological or surgical approach [70]. In any case, the patient must be stabilized before dystocia can be treated [59]:Medical therapy using oxytocin;Cloacal ovocentesis;Transcloacal cystoscopy ovocentesis [64,68,70] (Figure 13);A surgical approach involving a plastronotomy or prefemoral celiotomy [34].

### 5.3. Oophoritis

Oophoritis is acute or chronic inflammation of the ovary.

A case of granulomatous oophoritis was described in a specimen of *Chelydra serpentina*, which showed anorexia immediately after hibernation. Blood tests revealed pronounced heterophilia and monocytosis, and ovarian histology revealed the presence of cholesterol crystals associated with foamy macrophages and giant cells. During an oophorectomy, adhesions to the coelomic walls and diphtheroidal plaques on the surface were noted [37].

Oophoritis may be associated with other reproductive pathologies, such as preovulatory follicular stasis, dystocia, and yolk coelomitis. In such cases, follicular retention or rupture can promote bacterial proliferation and ovarian inflammation [91].

### 5.4. Salpingitis

Salpingitis is defined as acute or chronic inflammation of the oviduct.

#### 5.4.1. Causes

Common causes are chronic egg retention and backflow of fecal/urine material from the cloaca. Frequent causes of fecal/urinous ascent are dystocia linked to oversized eggs stuck in the pelvic canal. In Greek tortoise (*Testudo graeca*), egg retention has been described associated with *Proteus* and *Pseudomonas* spp. infection of the cloaca, which consequently led to ascending salpingitis [69]. A *Trachemys scripta elegans* was diagnosed with obstructive dystocia consequent to an egg stuck in the pelvic canal with associated bacterial salpingitis caused by *Escherichia coli* and *Proteus mirabilis* [67]. In another case, a 19-year-old specimen of *Siebenrockiella crassicollis* was diagnosed with bacterial salpingitis due to *Citrobacter freundii* following the breaking of an egg [37]. Granulomatous salpingitis caused by cardiovascular flukes that can carry eggs inside oviducts has also been reported [12].

#### 5.4.2. Clinical Signs

Clinical signs vary according to the cause and are usually infertility and/or dystocia. The eggs produced may be irregular.

#### 5.4.3. Diagnosis

By ultrasound or CT, it is possible to visualize the dilatation of the oviducts and their contents (often liquid); an intraluminal endoscopy and coelioscopy also provide a good contribution in confirming the diagnosis [87]. Microbiology via cloacal discharge or coelomic effusion may also suggest the presence of salpingitis, but it is important to remember that an oviduct has a bacterial microflora [51].

#### 5.4.4. Treatment

Therapy with antibiotics and/or anti-inflammatories may be effective in mild or recent cases but is unlikely to be successful in cases of chronic infection. The choice of proper antibiotic should be evaluated after the sensitivity test. When medical therapy fails, the elective surgical treatment is an ovariosalpingectomy.

### 5.5. Cloacitis

The term “cloacitis” refers to inflammation or infection of the cloaca.

#### 5.5.1. Causes

Cloacitis is a multifactorial pathology, and any damage to the cloacal region associated with environmental alterations can induce the excessive bacterial proliferation that characterizes it. The most common cause is repeated mating, which occurs especially when the sex ratio of the colony favors males (for which the F/M ratio should be 1/4–1/6). This phenomenon becomes more pronounced when the colony consists of specimens of *T. hermanni* due to the horny spur on the top of the male’s tail, which is used to harpoon the female during copulation. *Proteus mirabilis* and *Pseudomonas aeruginosa* can be opportunist bacteria during cloacitis in this species [92].

Other causes could include bite trauma, intestinal or urinary infections, constipation/diarrhoea, endo/ectoparasitosis (e.g., *Chelonacarus elongatus* is an acariasis typical of the cloacal wall of sea turtles that can cause cloacitis [12]), neurological deficits of the cloacal apparatus or sphincter, accumulation of urates, dystocia, alteration of the egg shell, alterations of the shell, neoplasms (e.g., fibropapillomas), and infectious pathologies (a study in 2022 diagnosed cloacal mycobacteriosis in a *Testudo hermanni* [34]).

Finally, it should be known that many management and/or food errors are considered predisposing factors.

#### 5.5.2. Clinical Signs

The animal is characterized by the presence of inflammation and laceration of the cloaca and pericloacal tissues, and is often associated with tenesmus [52].

Sometimes these lesions are very deep and are characterized by evident swelling, an abundant presence of necrotic tissue, and the formation of fistulous tracts. Often, cloacitis is complicated by myasis in the warm season.

#### 5.5.3. Diagnosis

The clinical presentation is diagnostic itself. When not related to trauma, secondary tests, such as swabs, blood tests, and imaging tests, sometimes help to determine the real cause of cloacitis.

#### 5.5.4. Treatment

Treatment for traumatic cloacitis is medical and generally involves topical and/or systemic therapies depending on the patient and is associated with management correction. Keeping the cloacal area as protected as possible, for example, with prevention from repeatedly crawling on the ground (e.g., by applying a small bump on the anal scutes), is recommended. Surgical therapy is usually characterized by curettage of the lesions.

The management of lesions must never disregard the assessment of possible consequences, such as dystocia [34].

### 5.6. Egg Yolk Coelomitis

Coelomitis is an inflammatory response of the coelom [54].

#### 5.6.1. Causes

It usually appears secondary to follicle rupture, salpingitis, oophoritis, and dystocia. Sometimes, the cause is iatrogenic, for example, after surgical procedures [54,60].

#### 5.6.2. Clinical Signs

Clinical signs are nonspecific and include lethargy, dysorexia, anorexia, diarrhea, decreased fecal and urate production, and pain upon coelomic palpation [66]. Lower respiratory tract disease, a distended coelom, dark discoloration of the skin, or death without premonitory signs have also been reported [59]. If septicemia occurs, diffuse erythema of the scutes or hemorrhagic petechiae can be observed [90].

#### 5.6.3. Diagnosis

Ultrasound, cystoscopy, and CT also allow for the evaluation of the disease and of coexisting and/or consequent pathological conditions, such as celomatic effusions [34].

Blood tests may be useful, and the presence of heterophilia, particularly in the presence of toxic heterophiles, may indicate a marked inflammatory and/or infectious component [34]. Sometimes there is evidence of multiple organ failure, renal dysfunction, or hypoglycemia [59].

Aspiration and analysis of the coelomic liquid allows for timely action via the introduction of targeted therapeutic protocols. The presence of basophils and yolk droplets of variable sizes can be detected in patients with egg coelomitis [32].

#### 5.6.4. Treatment

The therapeutic approach is only surgical and consists of removing the cause of coelomitis via an ovariectomy and salpingectomy. It is essential to thoroughly lavage the coelom and to obtain samples for histological and microbiological examinations to determine the correct therapy [1]. However, the prognosis remains confidential/poor.

### 5.7. Oviductal/Cloacal Prolapse

Prolapse is leakage of the oviduct through the cloaca, or the cloaca itself prolapses out of the cloacal orifice. This prolapse can be complete, i.e., involve both oviductal structures or only one of the two oviducts [93] (Figure 14 and Figure 15).

#### 5.7.1. Causes

The causes of this disease include generalized weakness; metabolic diseases, such as hypocalcemia, hyperestrogenism, and consequent cloacal hypertrophy; dehydration; obesity; parasitosis; neurological dysfunction; coelomic masses; overexertion (dyspnea, constipation, retained eggs, egg laying, and bladder stones); distal genital urinary/digestive tract infections; and treatments with oxytocin and oviductal neoplasia [94]. Often, prolapse occurs during an episode of dystocia or when the animal has partially laid eggs [83].

#### 5.7.2. Clinical Signs

At the clinical examination, the patient presents with a prolapsed organ through the cloacal opening. Depending on the severity of the prolapse itself, there could be an absence of symptoms up to severe dehydration and sepsis.

#### 5.7.3. Diagnosis

The morphology of the prolapse (invaginate, tubular, pedunculated, etc.) and its palpation provide important clues about the type of prolapsed organ (in this case, the differential diagnosis includes prolapse of the oviduct, rectum, bladder, and cloaca) [83]. Other clinical tests, such as diagnostic imaging, can certainly be useful for obtaining a correct diagnosis.

#### 5.7.4. Treatment

A prolapse represents an obstetrical emergency. If the prolapsed oviduct is still viable, conservative therapy can be administered, e.g., cleaning, reducing tissue oedema, and repositioning; if extensive necrosis or infection is present or the patient cannot be repositioned, the clinician can proceed with surgical therapy, e.g., removal. Before proceeding with this last option, the clinician should always try to understand the cause of the disease; however, depending on the diagnostic hypothesis, it is possible to proceed with combined celoma surgery (e.g., ovariectomy, cystotomy). Investigating the conditions that may predispose patients to/cause prolapse allows for the correction of these conditions, and therefore, the prevention of any recurrence [60].

### 5.8. Infertility

If copulation has occurred but no viable eggs are produced, the cause could be associated with male and/or female infertility disorders. It has been demonstrated that the mismanagement of animals is often the main cause. In females, however, infertility can be a consequence of salpingitis, oophoritis, cloacitis, follicular stasis, neoplasms of the reproductive system, or poor nutritional conditions [34]. Poor nutrition and absent/unsuitable oviposition sites are mentioned as causes [6].

#### 5.8.1. Clinical Signs

Infertility may manifest as an absence of egg production or the production of infertile eggs [1].

#### 5.8.2. Diagnosis

Recommended examinations of subjects consist of ultrasound examination of the reproductive tract, but blood tests, endoscopy, and tomography can also provide a more complete understanding of the specimen [6].

An analysis of the eggs is also important. Infertility must be differentiated from fertile eggs that fail to develop. Notably, unfertilized eggs may be produced and laid, but usually, only fertilized eggs are deposited [57]. A necroscopy or the microbiological testing of eggs are sometimes recommended. The clinician should not forget to ask about the parameters of incubation (such as humidity, temperature, or substrate composition) [1].

### 5.9. Neoplasia

Neoplasia is defined as abnormal or uncontrolled cell growth. Different tumors, such as teratomas (malignant and benign) [38,39,95], dysgerminomas [96,97], and leiomyomas [94,98], have been reported in the female reproductive tract of chelonians. Polyposis forms [98] and tumors of viral origin (fibropapillomas) can also occur and affect the cloacal/inguinal region.

#### 5.9.1. Causes

No causes have currently been identified, with the exception of fibropapillomatosis. In the latter case, marine chelonians are frequently affected by herpesvirus, and the recognized predisposing factors are the concomitant presence of parasites, bacteria, pollution, biotoxins, okadaic acid, adverse environmental conditions, and immunosuppression [12,99,100].

#### 5.9.2. Clinical Signs

The main clinical signs reported are lethargy [96], prolapse [94], anorexia [94,95,96], and lack of oviposition [39,95].

#### 5.9.3. Diagnosis and Treatment

Blood tests: In *Chelonia mydas*, fibropapillomatosis can be identified because it is associated with monocytosis [42], heterophilia, leukopenia [42], anemia [40,42], and hypoprotidemy [40]. A confirmatory diagnosis is made by means of collateral tests, such as diagnostic imaging and histology, and the principal treatment remains surgical [41].

### 5.10. Ovarian Torsion

A partial unilateral ovarian torsion, a condition rarely reported in reptiles [101,102,103], was diagnosed in a 14-year-old female Trachemys scripta presenting with inappetence and lethargy [104].

#### Diagnosis and Treatment

Diagnostic imaging revealed hyperechoic follicles, fluid accumulation, and absence of blood flow in a follicular group. An endoscopic ovariosalpingectomy confirmed a 360° follicular torsion with necrosis and coelomitis caused by *Klebsiella* spp. Post-surgical treatment led to full recovery, and the turtle was discharged after 15 days with a normal appetite.

## 6. Conclusions

Reproductive disorders in female chelonians represent a significant challenge in veterinary medicine, encompassing a wide range of conditions, such as infertility, follicular stasis, dystocia, oophoritis, salpingitis, cloacitis, and prolapse. These conditions are complex and require a nuanced approach to diagnosis and treatment. An in-depth understanding of the female reproductive anatomy and physiology is essential for identifying the most appropriate diagnostic and therapeutic strategies.

A critical component of effective management lies in the integration of species-specific knowledge, clinical history, and environmental management, which should form the cornerstone of any diagnostic process. Proper environmental factors, including photoperiod, temperature, humidity, nutrition, habitat, and sex ratios, have been highlighted as key contributors to reproductive health. This underscores the importance of educating owners and breeders on the optimal care practices to prevent and mitigate the onset of reproductive pathologies. Inadequate management, particularly in captive settings, can significantly contribute to the development of these disorders.

While extensive research exists on individual pathologies that affect chelonian reproduction, much of the available literature is based on isolated clinical cases or studies focused on specific species. This limitation highlights the need for further research to elucidate species-specific risk factors and to develop more standardized protocols for diagnosis and treatment across different chelonian species. Additionally, the lack of comprehensive studies that encompass diverse populations complicates the ability to draw generalized conclusions. Future studies should aim to address these gaps by focusing on larger sample sizes and cross-species comparisons.

In summary, reproductive disorders in chelonians are multifactorial and complex, requiring a holistic approach that combines accurate diagnosis, tailored treatment, and preventative measures based on a thorough understanding of the species and environmental factors. Only through continued research and clinical experience can we hope to develop effective and standardized management strategies to improve the reproductive health of these fascinating reptiles.

## Figures and Tables

**Figure 1 animals-15-01275-f001:**
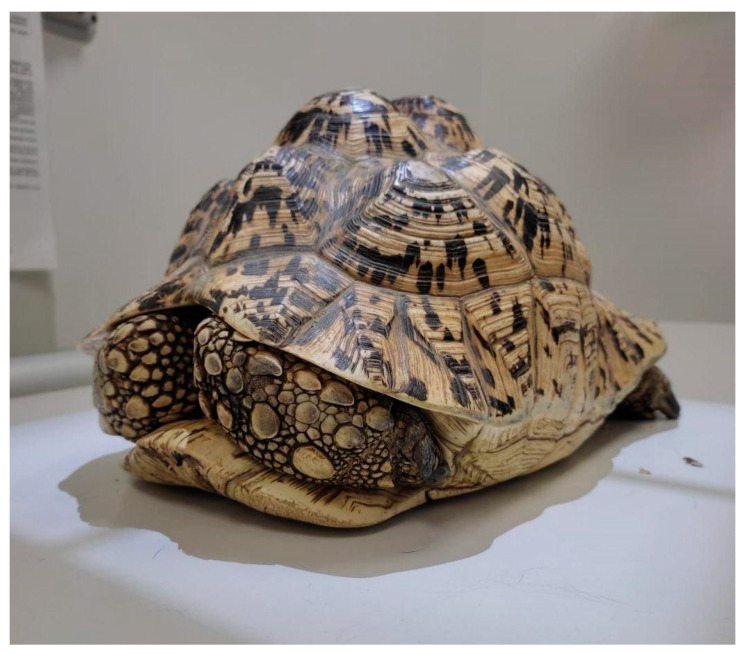
Positioning of a Stigmochelys pardalis tortoise in sternal recumbency on a dorsoventral radiograph.

**Figure 2 animals-15-01275-f002:**
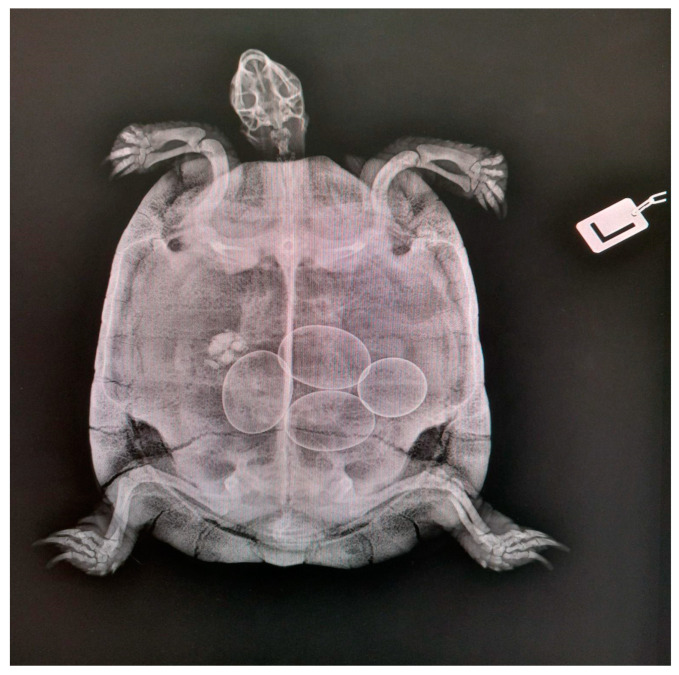
Dorsoventral radiograph of a Hermann’s tortoise. There are four eggs that appear normally mineralized and normally formed (courtesy of Ospedale Universitario Veterinario di Lodi).

**Figure 3 animals-15-01275-f003:**
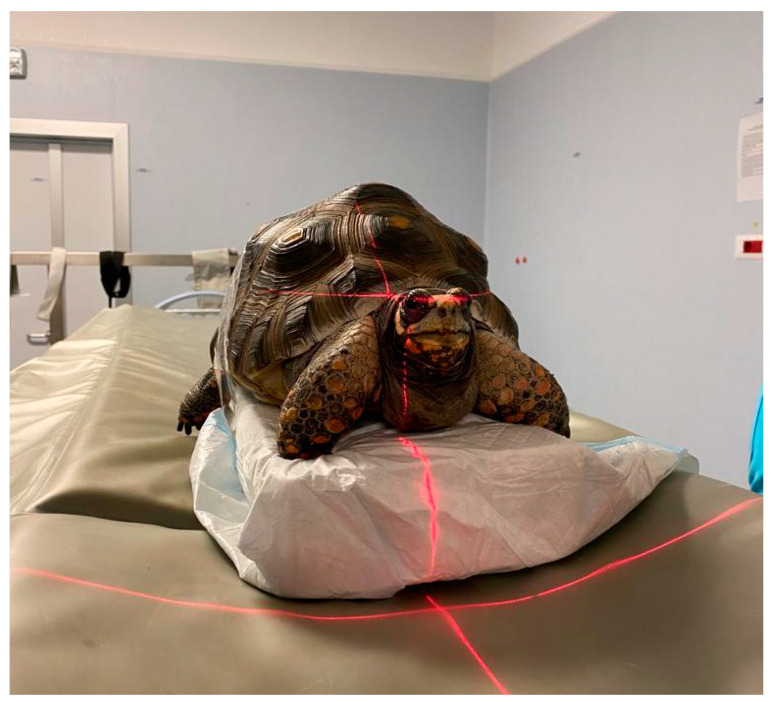
Positioning of a *Chelonoidis carbonarius* tortoise in sternal recumbency on a block for a TC scan.

**Figure 4 animals-15-01275-f004:**
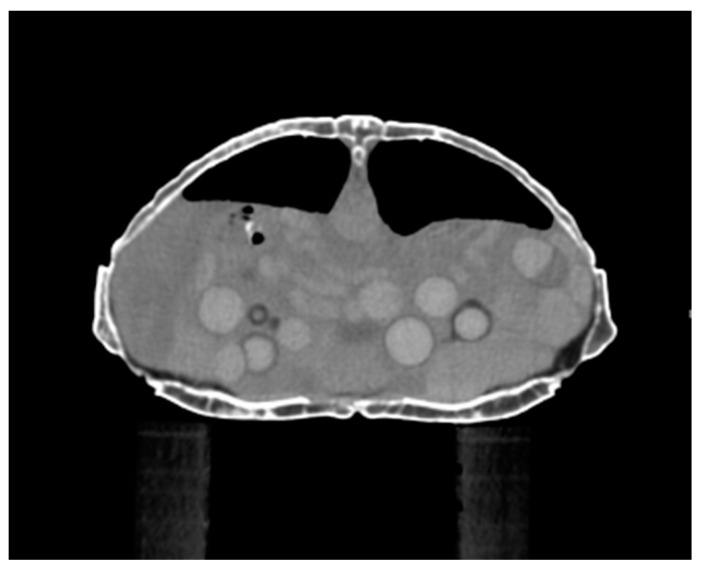
Transverse CT view of the middle of the coelomic cavity in *Chelonoidis carbonarius*. Multiple follicles in different stages are present (courtesy of Ospedale Universitario Veterinario di Lodi).

**Figure 5 animals-15-01275-f005:**
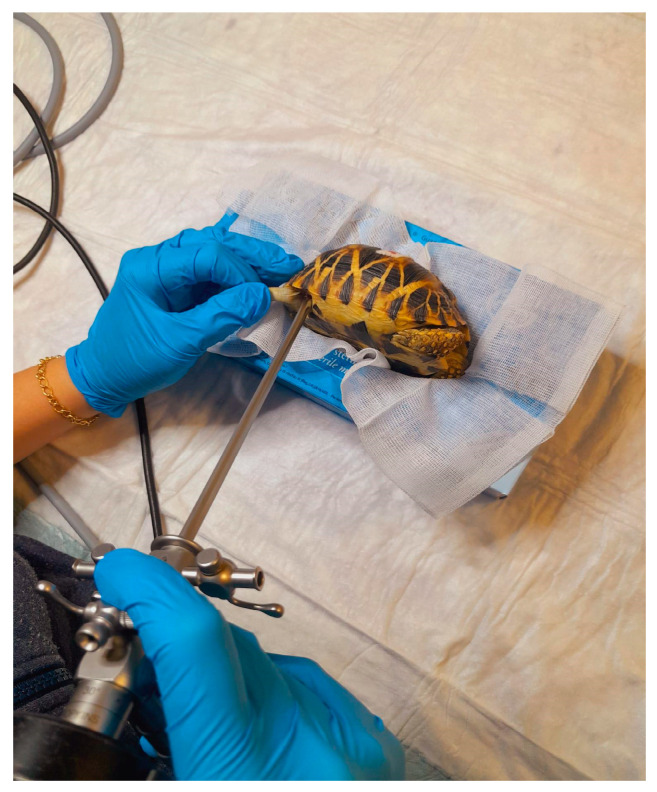
Tortoise in lateral recumbency for coelioscopy.

**Figure 6 animals-15-01275-f006:**
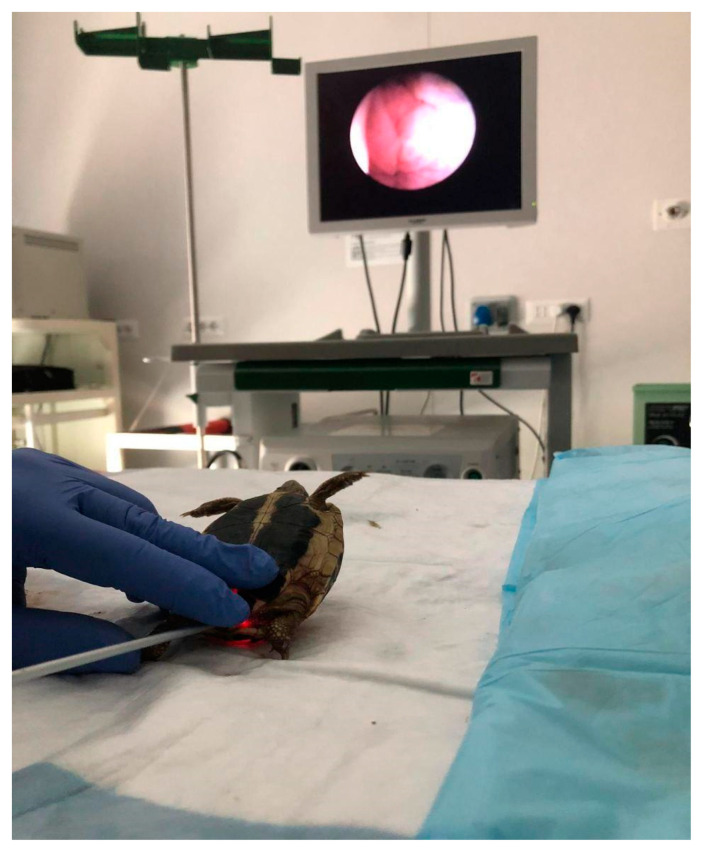
Tortoise in dorsal recumbency for cloacoscopy.

**Figure 7 animals-15-01275-f007:**
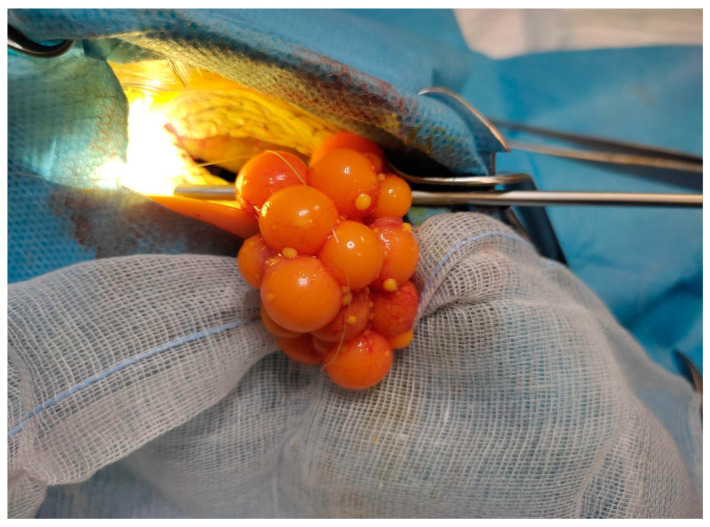
Prefemoral surgical approach to the chelonian coelom (*Trachemys scripta*). The ovary with follicles is completely exteriorized.

**Figure 8 animals-15-01275-f008:**
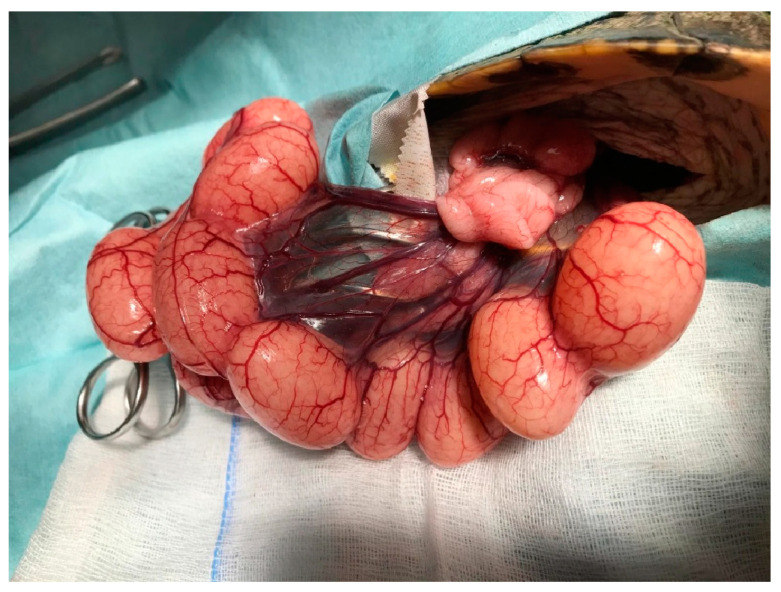
Prefemoral surgical approach to the chelonian coelom (*Trachemys scripta*). The oviduct is exposed and it is possible to see eggs due to the transparency.

**Figure 9 animals-15-01275-f009:**
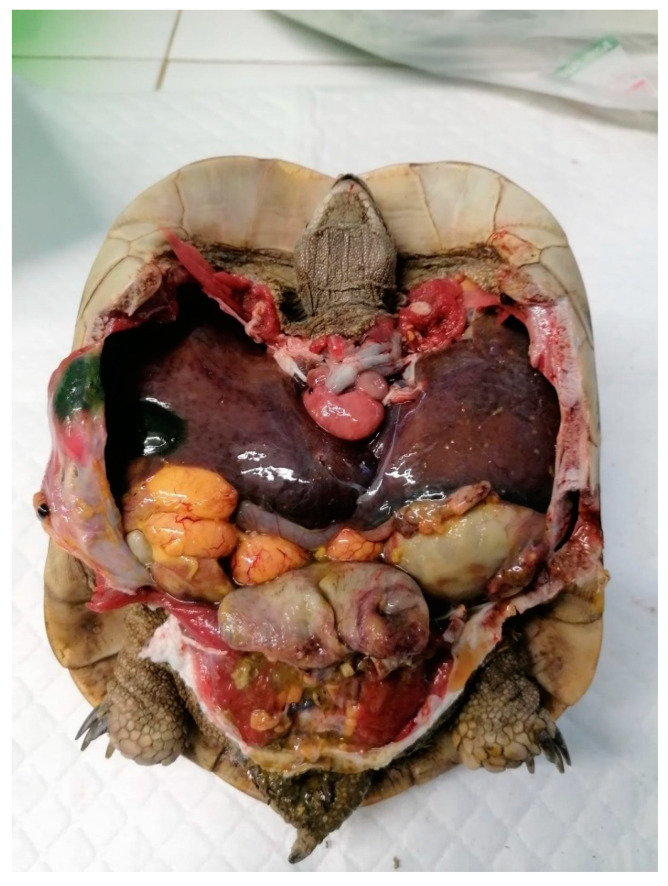
Eggs coelomitis associated with necrotic cystitis visible during necroscopy.

**Figure 10 animals-15-01275-f010:**
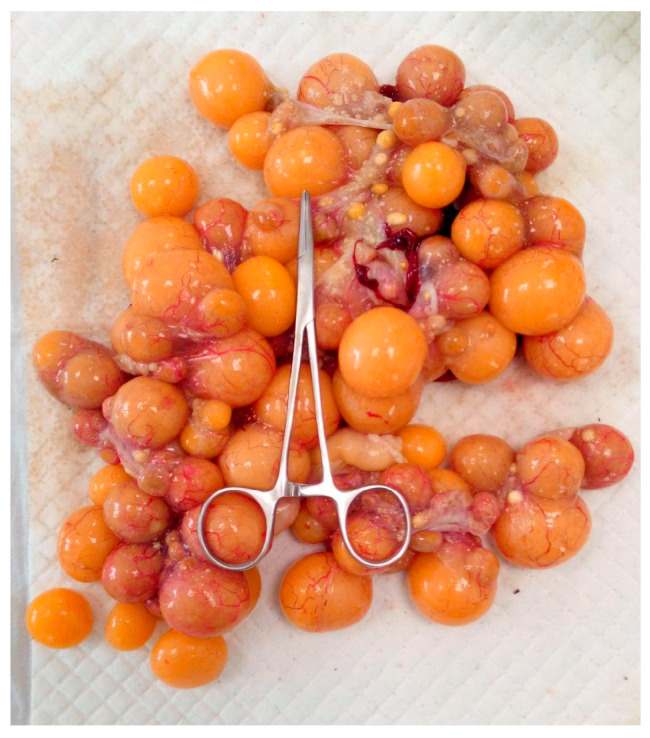
Ovary with multiple follicles in a mature turtle after resection.

**Figure 11 animals-15-01275-f011:**
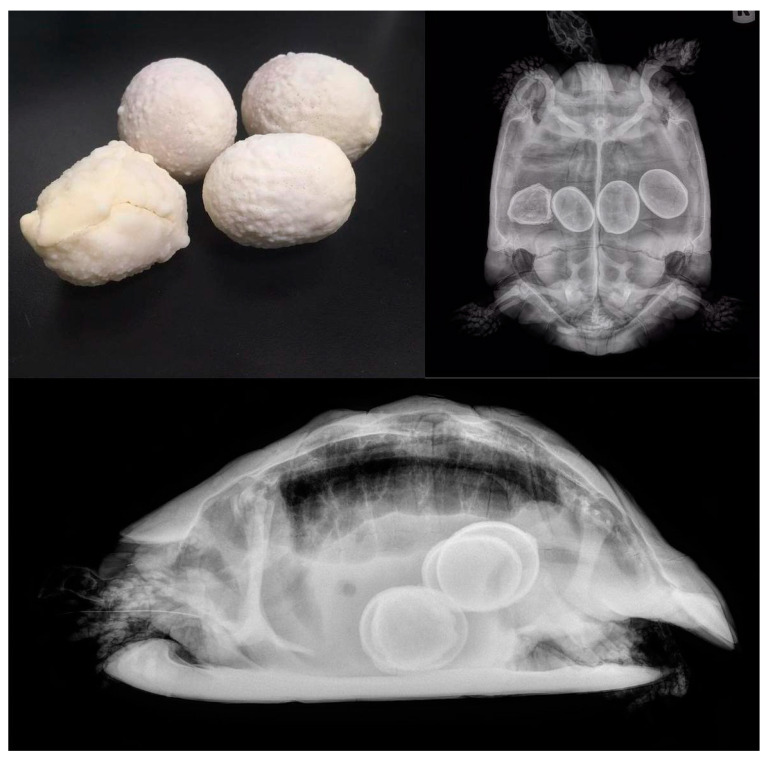
Chronic egg retention visible by radiography and after removal. All of them have thick and irregular walls caused by chronic retention in oviduct.

**Figure 12 animals-15-01275-f012:**
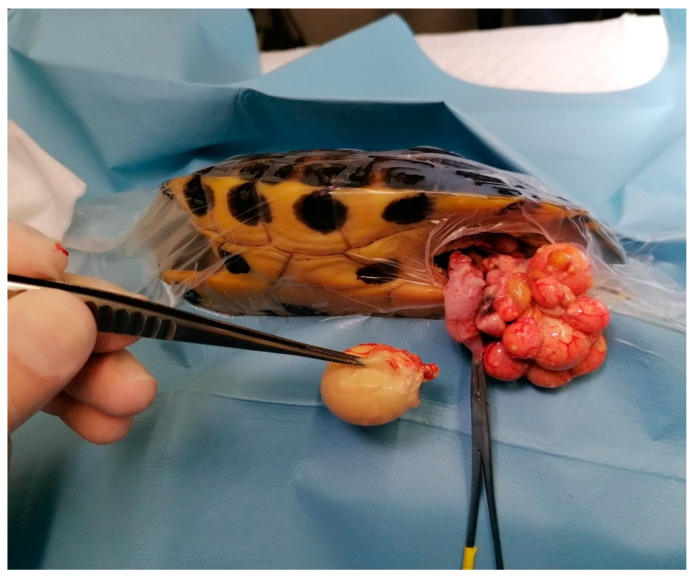
Ectopic egg after rupture of oviduct, which is visible during a prefemoral surgical approach, and consequent follicular stasis.

**Figure 13 animals-15-01275-f013:**
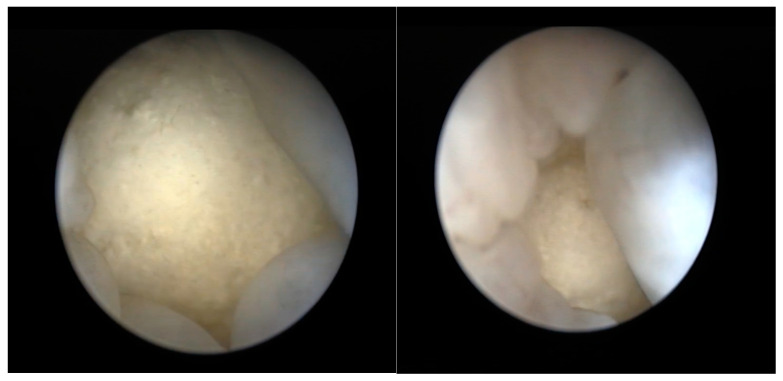
Ectopic egg in the urinary bladder.

**Figure 14 animals-15-01275-f014:**
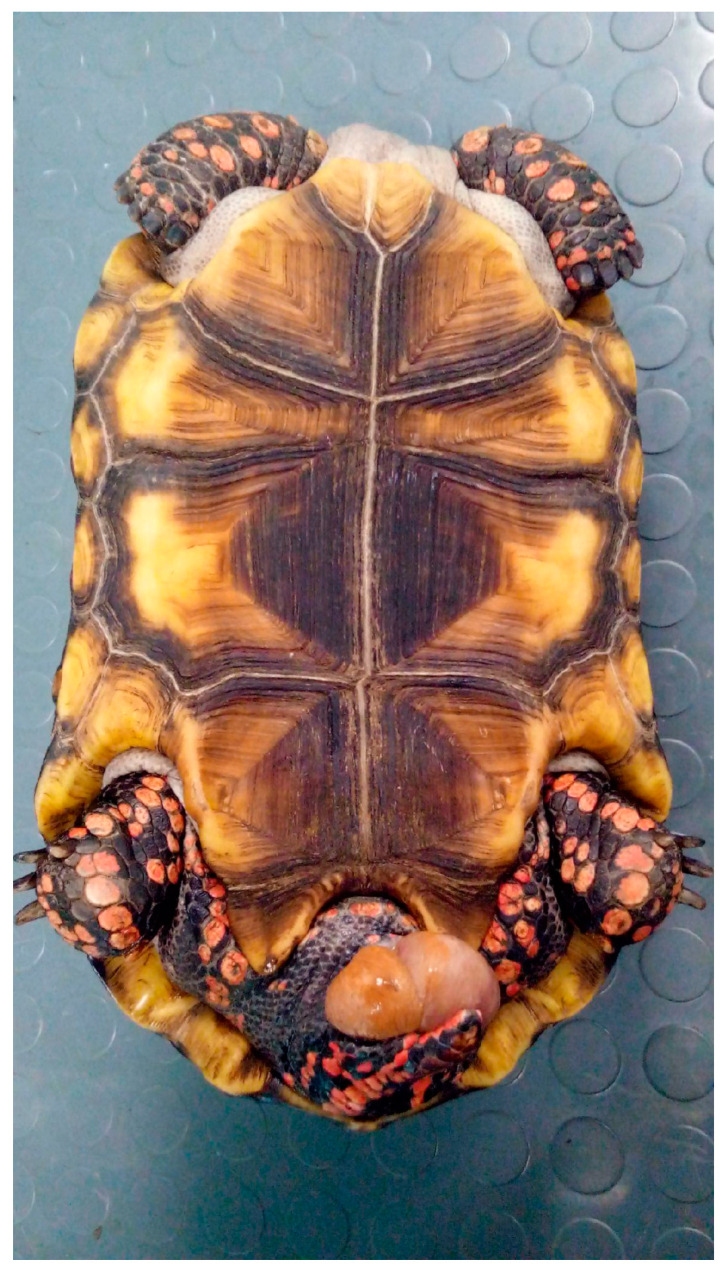
Cloacal prolapse in *Chelonoidis carbonarius*.

**Figure 15 animals-15-01275-f015:**
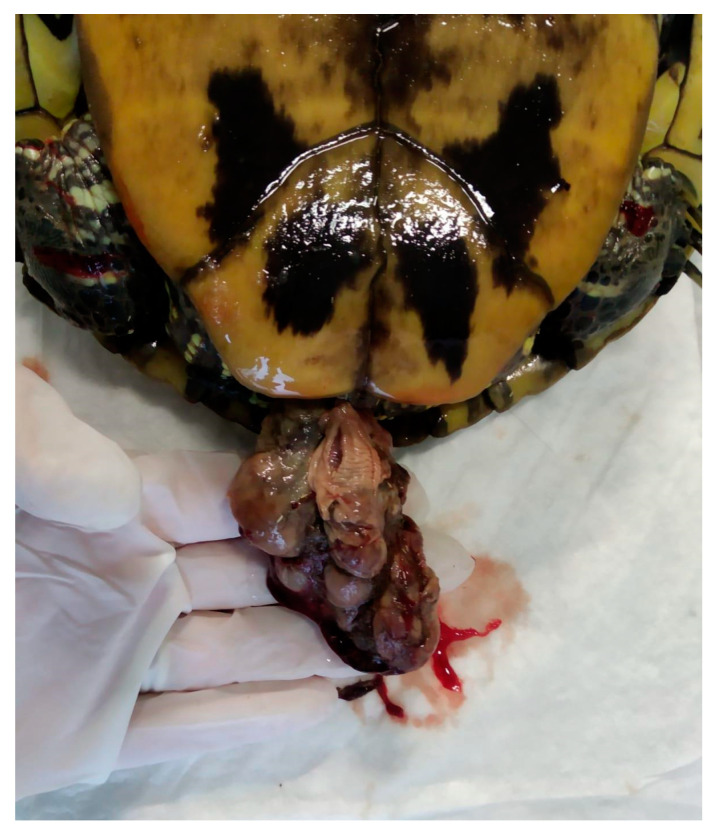
Oviductal prolapse *Trachemys scripta*.

## Data Availability

No new data were created or analyzed in this study.

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
