# Peer review of "Disorders of the Female Reproductive Tract in Chelonians: A Review"

_animals, 2025, doi:10.3390/ani15091275_

Round 1
Reviewer 1 Report
Comments and Suggestions for Authors
The structure of the text is divided into sections that address the anatomy, reproductive physiology, diseases, and treatments related to the female reproductive tract of chelonians. This facilitates the reader's understanding and follow-up. The authors also provide descriptions of pathological conditions, as well as diagnostic and therapeutic procedures. However, the review does not mention the use of a systematic methodology, such as PRISMA, for the selection and analysis of the included studies.
The PRISMA method is important for structuring systematic reviews in a clear and transparent manner. The absence of this method may indicate that the selection and analysis of studies did not follow a rigorous protocol, which can compromise the validity and reliability of the conclusions. This method also helps ensure that the selection of studies is comprehensive and unbiased. Without it, there may be a lack of clear criteria for the inclusion and exclusion of studies, which can result in selection bias.
Authors can consult: The PRISMA 2020 statement: an updated guideline for reporting systematic reviews. http://dx.doi.org/10.1136/bmj.n71
Another important aspect is the absence of a critical discussion on the points and articles addressed, which would make the manuscript more robust. Despite the high importance of the topic proposed by the authors, an adjustment of the text is fundamental for publication.
The paper is of great importance for chelonian medicine, and I accept the manuscript for publication as a review. Nonetheless, I emphasize that a systematic review using the PRISMA methodology would be very important to identify, analyze, and report aggregated evidence on reproductive disorders.
Comments on the Quality of English Language
I am not qualified to judge the quality of English writing.
Author Response
Thank you for your suggestion regarding the use of the PRISMA method. Following your recommendation, we have thoroughly revised the manuscript in order to align it as closely as possible with the PRISMA guidelines. Although this field is still developing and not all data meet systematic review standards, we believe that adopting the PRISMA approach improves the structure, transparency, and reproducibility of our review. We have therefore clarified our inclusion criteria, data sources, and the selection process for the literature, aiming to enhance the scientific rigor and clinical relevance of the article.
Reviewer 2 Report
Comments and Suggestions for Authors
This review article provides a description of the various disorders that can be encountered in (captive) chelonians. Although this review provides information that is pertinent to the veterinary practitioner that sees chelonian patients, the information is limited/concise and general. In its present form, the review is more written and structured as a summarized book chapter than as a review article. The scientific level of the content but especially of the writing should be increased if the authors wish to revise/resubmit the paper.
Some more specific remarks that may help to improve the manuscript:
Simple summary/Abstract:
L12:
‘The frequently encountered disorders are...’ suggest to rephrase (grammar). If these are the frequently encountered ones, the authors should also list and discuss more unusual disorders. Is it correct to state (based on the currently available literature) that reproductive neoplasms, follicular stasis, oophoritis, infertility... are frequently observed in chelonians, also in comparison to squamates? The prevalence and occurrence should be elaborated on in the discussion section.
L14: the selected pathology? Does this refer to reproductive disorders in chelonians in general? Also the reproductive biology of the species should be taken into account as written at L18?
‘Tortoise disease, chelonian’ seem to be inappropriate keywords. Also check appropriate use of capittals.
Introduction
Please add a paragraph about the prevalence of the various disorders that you will discuss, also in relation to other reptile taxa. Which pathologies are more common than others and why.
L 34 Please verify that ‘health check-ups’ is correct and the most appropriate wording
L35-37: unclear what the authors try to state here
L38: who else should they rely on the diagnose and treat their animals?
L39-41: the reference only deals with chelonians, this paragraph seems to relate to pets in general. Please reword entire paragraph.
L41: so secondary reproductive disorders are not discussed?
L42-43: the authors did not establish all the procedures, they summarize the information that is available in the literature
L58: here and in other places, remove striked words from the text
L78: ‘lethargy’?, L94: of course calcified eggs will first remain in the oviduct (check use of capital letters)
L83: ref 12 in correct place? How does this info about T. venusta relate to what the reader should know about this topic in general? Is nothing described in other species?
Section 1.2 (Suggest to add ‘Female’ to subtitle): also see previous remarks, this is a very concise section to describe the reproductive physiology and endocrinology in female chelonians, that is relatively incomplete and written in a confusing way. The authors should make an effort to improve this section and write it more complete and scientific. L95, reference 2,4: do these references cover the load? Do they really describe the retention of eggs or do they in turn refer to specific papers? Check if ref 4 is relevant here.
L106-107: isn’t this relevant to any disease problem that it is best to know the species and its natural needs and behavior?
L112-114: see remark higher (primary disorders in abstract).
L111-114: This is very vague. The authors should rewrite this and be more specific, what primary conditions outside of the reproductive tract can eventually lead to pathology that is discussed in this review. Maybe it is better to already mention this in the intro. The authors should attempt to provide a better overview of what the clinical exam should constitute and anamnesis and management assessment should be elaborated on first. What are the specific points that should be addressed here (from factors leading to metabolic diseases (diet, housing,...) to assessing behavior that could indicate the presence of a reproductive problem).
Section 2.1 Collateral tests is ill-written. The authors are providing general definitions of the different types of collateral tests. The latter is not in place here. The information is fragmented and summing up facts rather than providing a coherent section. Suggest to rewrite the entire section in a structured way, emphasizing the need for a multidirectional approach and tailor it more specific to reproductive disorders in chelonians. Illustrate how the different diagnostic modalities can complement each other. The part about imaging seems to be too concise, more information (practicalities and clinical importance/relevance) would be in place here.
L275: ‘new approach’; this approach is used for quite some time... Referencing here and in other places is poor. Other references included in you citations should be included here, especially the ones that really describe the method or compare the different surgical approaches. Could you please provide clear proof from the literature that complications are more common following plastronectomy?
I would suggest delete the description of the surgical techniques as these procedures have been described in detail elsewhere, and the description here is a rather a summary that does not provide the necessary details. In addition, the phrasing is poor. Suggest to replace this section by a providing an overview of the therapeutic approach to reproductive disorders in female chelonians and what criteria should be used in decision taking.
Fig 10. Why was the oviduct ruptured in case of follicular stasis?
L341 “Diseases of the reproductive system”, replace by “Disorders of the female chelonian reproductive system” for consistency and clarity (you are not discussing the male reproductive system. Please also check in other places.
In my opinion, ectopic eggs should not be discussed as a separate disorder but rather as a form or complication of dystocia. L344: or most frequently? Clarify. L346: but also developmental disorders play role in this and other factors that prevent the eggs to be evacuated via the cloacal vent?
5.1.3. and 5.1.4: as written now, the text has the format of bullet points on a slide presentation. Please write in full sentences here and for the other disorders that are discussed. 5.1.4: as mentioned above, it is important for the reader to know when which therapeutic strategy should be used and this can be provided higher (see previous comment)
L376: “an accumulation of follicles inside the ovary”, this does not seem to be a correct, scientifically sound definition. Please provide a sound scientific definition and relate it to the occurrence/pathogenesis of this disorder in chelonians. What do the other mean with concomitant pathologies? Which systemic pathologies? Is there scientific proof of the endocrinological basis? If e.g. oophoritis or neoplasia is present, does it still make sense to call the disorder follicular stasis, please provide a referenced statement about this.
L416 might consider to replace terms as ‘mother’ and ‘pregnancy’
L418: ectopic eggs confirms this is included under dystocia
5.3.1 this section is important but the phrasing is awkward and should be largely improved
L431: Clinical signs appear when...: what does this mean? The scientific value of sentences like this is absent.
5.3.3 the order of modalities is confusing, e.g. ‘cystoscopi’ mentioned first at L453
5.4: any possible microbial causes described in chelonians (or refer to other taxa)?
L477: please always provide the common and scientific name (first mentionà as is universally done in scientific papers. Apply throughout the manuscript.
L488: does explorative coelioscopy has value here (and maybe also in case of other disorders) towards establishing a diagnosis
5.6 not sure if cloacitis should be included as a reproductive disorder (especially as described here)
5.7 Why do the authors suddenly not add the causes section? Please add.
5.8 please delete this section. It is inacceptable to suddenly shift from listing clinical entities to providing an overview of known infections that might infect the rproductive tract. Insert them in other places, if appropriate. 5.8.3 is the best illustration that this section should be deleted
L635: it seems somewhat awkward to state that no causes have been identified taking into account the basic definition of ‘neoplasia’
Please improve the wording of the conclusions section, especially the last paragraph is very unclear
In various references, not all authors are listed. Please carefully check.
Comments on the Quality of English LanguageSee other comments
Author Response
Thank you very much for your helpful suggestions. We have revised the entire manuscript according to your recommendations, except for the points detailed in the attached file. We truly appreciate your corrections and constructive feedback.

Round 2
Reviewer 1 Report
Comments and Suggestions for Authors
The authors significantly improved the final manuscript following the reviewers' considerations. As a result, this review is an important contribution for professionals working with reptile clinical practice and surgery.
Reviewer 2 Report
Comments and Suggestions for Authors
The authors have made an effort to address my comments. Quality of English language can still be improved. In my opinion, section 2.1 is still largely redundant as written now and could be replaced to providing a more general diagnostic approach to reproductive disorders in chelonians.
Comments on the Quality of English LanguageThe quality of the English language can still be improved. Just an example: "Reproductive tract disease is generally detected in blood, coelomic and cloacal fluids and tumours. Depending on the situation, it is possible to obtain samples with swabs, fine needle aspiration or tissue imprinting, and..."